# BRCAness as an Important Prognostic Marker in Patients with Triple-Negative Breast Cancer Treated with Neoadjuvant Chemotherapy: A Multicenter Retrospective Study

**DOI:** 10.3390/diagnostics10020119

**Published:** 2020-02-21

**Authors:** Yoshimasa Kosaka, Yutaka Yamamoto, Hirokazu Tanino, Hiroshi Nishimiya, Mutsuko Yamamoto-Ibusuki, Yuko Hirota, Hirotaka Iwase, Seigo Nakamura, Sadako Akashi-Tanaka

**Affiliations:** 1Department of Breast and Endocrine Surgery, Kitasato University School of Medicine, Sagamihara 252-0374, Japan; y-kosaka@med.kitasato-u.ac.jp (Y.K.);; 2Department of Breast and Endocrine Surgery, Graduate School of Medical Sciences Kumamoto University, Kumamoto 860-8556, Japan; 3Division of Breast Surgery, Department of Surgery, Kobe University Graduate School of Medicine, Kobe 650-0017, Japan; 4Department of Molecular-Targeting Therapy for Breast Cancer, Kumamoto University Hospital, Kumamoto 860-8556, Japan; 5Department of Diagnostic Pathology, Showa University Koutou Toyosu Hospital, Koutou 135-8577, Japan; 6Department of Breast Surgical Oncology, Showa University School of Medicine, Shinagawa 142-8666, Japan

**Keywords:** breast cancer, BRCAness, triple-negative breast cancer, neoadjuvant chemotherapy, prognosis

## Abstract

Triple-negative breast cancer (TNBC) has several subtypes. The identification of markers associated with recurrence and poor prognosis in patients with TNBC is urgently needed. BRCAness is a set of traits in which *BRCA1* dysfunction, arising from gene mutation, methylation, or deletion, results in DNA repair deficiency. In the current study, we evaluated the clinical significance and prognosis of BRCAness in a multicenter retrospective study. Ninety-four patients with TNBC treated with neoadjuvant chemotherapy were enrolled from three university hospitals for this retrospective study. BRCAness was evaluated in 94 core needle biopsy (CNB) specimens prior to neoadjuvant chemotherapy and 49 surgical specimens without pathological complete response (pCR). The samples were assessed using multiplex ligation-dependent probe amplification, and the amplicons were scored. Of the 94 patients, 51 had BRCAness in CNB specimens. There were no significant differences in pCR rates or recurrence between the BRCAness and non-BRCAness groups. Among surgical specimens, the BRCAness group had a significantly shorter recurrence-free survival and overall survival compared with the non-BRCAness group. The BRCAness of surgical specimens was found to be an important marker to predict prognosis in patients with TNBC after neoadjuvant chemotherapy. A clinical trial to assess the clinical impact of carboplatin with BRCAness is planned.

## 1. Introduction

Breast cancer is the most frequently diagnosed cancer and the leading cause of cancer-related death in women worldwide [1,2]. Breast cancer can be classified into at least five intrinsic subtypes based on gene expression profiling [3,4,5,6]. Triple-negative breast cancer (TNBC) is one of these subtypes and is defined as estrogen receptor, progesterone receptor, and human epidermal growth factor receptor 2 (HER2) negative by immunohistochemistry. TNBC is associated with poor long-term outcomes compared with other breast cancer subtypes. However, approximately one-third of patients with TNBC who achieve pathological complete response (pCR) have a good prognosis following neoadjuvant chemotherapy [7,8]. Recently, TNBC was classified into six phenotypes, i.e., basal-like1, basal-like2, immunomodulatory, mesenchymal, mesenchymal stem-like, and luminal androgen receptor, by gene profiling [9]. However, this gene profiling technique cannot identify therapeutic agents.

*BRCA1* and *BRCA2* mutations are relatively frequent in patients who have a family history of cancer, i.e., hereditary breast and ovarian cancer. TNBC is strongly correlated with *BRCA1/2* mutation status, and up to 20% of patients with TNBC are carriers of these mutations [10]. The *BRCA1* and *BRCA2* genes encode proteins involved in double-stranded DNA break repair; thus, *BRCA* mutation-associated cancers may be more sensitive to chemotherapeutic agents that cause DNA damage, such as platinum-based agents [11,12,13,14]. “BRCAness” refers to some sporadic cancers that share phenotypic characteristics with tumors carrying *BRCA1/2* mutations, such as methylation of *BRCA1/2* promoters and low *BRCA1* gene expression [15]. Recently, multiplex ligation-dependent probe amplification (MLPA) assays were developed to determine the BRCAness classification of breast tumors. Tumors classified into this category were proposed to behave similarly to *BRCA*-mutated cancers in terms of natural history and response to systemic therapy.

In this study, we investigated whether BRCAness was associated with the pCR rate after neoadjuvant chemotherapy and whether this classification affected recurrence-free survival (RFS) and overall survival (OS) rates in a multicenter study.

## 2. Materials and Methods

### 2.1. Patients

All of the ninety-four patients with stage I–III TNBC, who were diagnosed and treated with neoadjuvant chemotherapy at Kitasato University Hospital, Showa University Hospital, and Kumamoto University Hospital, were enrolled between January 2005 and March 2015. The median observation period was 32 months (range, 4–119 months). The average age of patients was 51.3 years (range, 24–74 years). Seven of the patients were classified as clinical stage I, 64 patients were classified as stage II, and 23 patients were classified as stage III. The patients received the following regimens: anthracycline regimens which, given every 21 days, were 4 cycles of FEC (5FU 500 mg/m^2^, epirubicin 100 mg/m^2^, and cyclophosphamide 500 mg/m^2^), 4 cycles of EC (epirubicin 90 mg/m^2^ and cyclophosphamide 600 mg/m^2^), or 4 cycles of AC (doxorubicin 60 mg/m^2^ and cyclophosphamide 600 mg/m^2^). Taxane regimens were 4 cycles of docetaxel (75 mg/m^2^, every 21 days), or 12 cycles of paclitaxel (80 mg/m^2^, every 7 days). No platinum salts were used for this study because it is not covered by Japanese insurance. Anthracyclines followed by taxanes were used in 86 patients for the neoadjuvant regimen. The efficacy of neoadjuvant chemotherapy was determined in terms of the pCR rate, which was defined as ypT0/Tis/N0; pCR was observed in 45 patients (47.9%). Twenty-two patients experienced recurrence after surgery, and 14 patients died from breast cancer. These clinicopathological data were originally collected from a database or medical records at each hospital. The characteristics of the 94 patients are shown in Table 1.

### 2.2. DNA Isolation and MLPA

We followed the methods of Tanino et al. as below [16]. Core needle biopsy (CNB) specimens before neoadjuvant chemotherapy and surgical specimens of non-pCR were used for MLPA analysis. Representative hematoxylin and eosin-stained slides from formalin-fixed, paraffin-embedded (FFPE) specimens were reviewed by a pathologist. Tumor tissues were selected and dissected using a scalpel. DNA was isolated from tumor tissue using a QIAamp DNA FFPE Tissue Kit (Qiagen, Hilden, Germany).

The classification of subtypes of BRCAness was performed using MLPA with P376-B2 BRCAness probemix (MRC-Holland, Amsterdam, the Netherlands) as previously reported [17]. This probemix covers the chromosomal regions that have been found to be gained in 3q22-29, 6p21-22, 10p14, 12p13, and 13q31-34 and lost in 3p21, 5q12-23, 10q23, 12q21-23, 14q22-24, and 15q15-21 in previous studies [12]. MLPA was carried out at FALCO Biosystems Ltd. and was performed according to the manufacturer’s instructions. Briefly, 5 μL DNA (50–100 ng) was denatured at 98 °C for 5 min and subsequently cooled down to 25 °C. After adding the probe mix, the sample was denatured at 95 °C for 1 min, and the probes were allowed to hybridize at 60 °C for 16 h. Probe ligation was performed with temperature-stable ligase-65 for 15 min at 54 °C. The ligase was then inactivated by incubation at 98 °C for 5 min. Polymerase chain reaction (PCR) was carried out by 35 cycles at 95 °C for 30 s, 60 °C for 30 s, and 72 °C for 60 s, followed by a final extension at 72 °C for 20 min. The PCR products were analyzed on a 3130 × l genetic analyzer (Life Technologies, Foster City, CA, USA) using Genescan 500 ROX size standards (Life Technologies, Foster City, CA, USA). Data analysis was performed using Coffalyser.NET software (MRC-Holland). The relative copy number ratio for each sample was compared with human genomic DNA (Promega, Madison, WI, USA) as a reference sample using Coffalyser.NET default settings. The BRCAness score was calculated according to the relative copy number ratios of various DNA sequences. The relative copy number ratios from Coffalyser.NET for all 38 target-specific probes were used for prediction analysis for microarrays (PAM). The training set generated by MRC-Holland with P376-B2 Lot 0911 was used for the PAM. Each sample was analyzed twice, and the average score was used for this analysis. The cutoff value for defining BRCAness was 0.5. Validation of the BRCAness assay regarding *BRCA1/2* mutation and *BRCA* promoter methylation was performed in the previous study and is guaranteed by FALCO Biosystems [12,18].

### 2.3. Data Analysis

The patients were classified into the BRCAness group or non-BRCAness group. Clinicopathological factors, clinical efficacy of neoadjuvant chemotherapy, pCR rates, recurrence, and survival were compared between the two groups. TNM classification was defined based on the seventh edition of the Union for International Cancer Control.

### 2.4. Statistical Analysis

The significance of the differences between the BRCAness and non-BRCAness groups was assessed using t-tests and Chi-square tests for clinicopathological variables. RFS and OS were calculated using the Kaplan–Meier method, and survival differences were assessed using log-rank tests. Results with *p* values of less than 0.05 were considered to indicate statistical significance. All statistical analyses were conducted with the SAS software package (JMP, SAS Institute, Cary, NC, USA).

### 2.5. Statement of Ethics

This study was performed according to the guidelines of the Declaration of Helsinki, as amended in Edinburgh, Scotland in October 2000. Institutional Review Board approval and written informed consent were obtained from all patients. The study was approved by the ethics committee of each hospital (the approval code: B15-161, the approval date: 25 April 2016) as follows: Institutional Review Board for Human Genome Research of Kitasato University, Institutional Review Board of Showa University, and Ethics Committee for clinical research & advanced medical technology at the Faculty of Life Sciences, Kumamoto University.

## 3. Results

### 3.1. RFS and OS of all Patients

At the median follow-up of 32 months, 22 RFS events and 14 OS events had been registered. The five-year RFS rate was 73.4% (Figure 1a). The five-year OS rate was 78.7% (Figure 1b).

### 3.2. BRCAness of CNB Specimens and Clinicopathological Factors

Of the 94 patients with TNBC, 51 patients (54.3%) had BRCAness, and 43 patients (45.7%) did not have BRCAness (non-BRCAness) in CNB specimens. We evaluated BRCAness and clinicopathological factors, such as age, cT (cT1-cT2 versus cT3-cT4), cN (cN0 versus cN1-cN3), cStage, and response to neoadjuvant chemotherapy (pCR versus non-pCR). No statistically significant differences were observed between the BRCAness and non-BRCAness groups with regard to these clinicopathological factors (Table 2).

### 3.3. BRCAness of Surgical Specimens and Clinicopathological Factors

Of the 49 patients with non-pCR by neoadjuvant chemotherapy, 19 patients (38.8%) had BRCAness, and 30 patients (61.2%) did not have BRCAness in surgical specimens. In the clinicopathological analysis, patients in the BRCAness group were significantly younger than those in the non-BRCAness group (mean age 47.0 versus 53.5 years, respectively; *p* < 0.05). Significantly-increased recurrence was observed in the BRCAness group compared with that in the non-BRCAness group (68.4% versus 30.0%, respectively; *p* < 0.05) after neoadjuvant chemotherapy. No statistically significant differences were observed between the BRCAness and non-BRCAness groups with regard to cT, cN, and cStage (Table 3).

### 3.4. BRCAness of CNB Specimens and RFS/OS

At the median follow-up of 32 months, 22 RFS events and 14 OS events had been registered. There were no significant differences between the BRCAness and non-BRCAness groups in terms of five-year RFS rate (68.4% versus 80.2%, respectively; *p* = 0.16; Figure 2a) and five-year OS rate (76.6% versus 82.3%, respectively; *p* = 0.19; Figure 2b). 

### 3.5. BRCAness of Surgical Specimens and RFS/OS

The five-year RFS rate in the BRCAness group was significantly lower than that in the non-BRCAness group (23.1% versus 66.7%, respectively; *p* < 0.01; Figure 3a). Moreover, the five-year OS rate in the BRCAness group was significantly lower than that in the non-BRCAness group (47.2% versus 67.2%, respectively; *p* < 0.05; Figure 3b).

## 4. Discussion

This is the first report showing RFS and OS according to BRCAness in patients with TNBC treated with neoadjuvant chemotherapy. The results of this study suggested that the BRCAness of surgical specimens after neoadjuvant chemotherapy of TNBC was an important marker for predicting recurrence or poor prognosis. In surgical specimens, the BRCAness group was associated with high rates of recurrence (68.4%), similar to the results of our previous study (7/9 patients, 77.8%). However, our previous study was thought to be underpowered because the data were from a single institution and only included 40 patients and a retrospective study [16].Therefore, we planned this pooled analysis to avoid a patient’s bias as much as possible. From this pooled analysis, we demonstrated that BRCAness testing using surgical specimens after neoadjuvant chemotherapy was a strong predictive marker of RFS and OS. Based on these results, additional therapies for patients who did not achieve pCR after neoadjuvant chemotherapy, as in the CREATE-X trial (adjuvant capecitabine treatment) and the KATHERINE trial (adjuvant trastuzumab emtansine treatment in HER2-positive patients), will be needed for patients showing BRCAness; thus, we have planned a new clinical trial to evaluate this [19,20].

Notably, in CNB specimens, we did not find any correlations between BRCAness and pCR rates after neoadjuvant chemotherapy. However, patients in the non-BRCAness group tended to more frequently achieve pCR than patients in the BRCAness group (58.1% versus 39.2%, *p* < 0.1). When we defined the cutoff value as 0.4, based on a study by Akashi-Tanaka [21], the non-BRCAness group had a significantly higher pCR rate than the BRCAness group (61.5% versus 38.2%, *p* < 0.05). This result was consistent with Akashi-Tanaka’s study. Regardless of the results of BRCAness analysis, TNBC still needed chemotherapy in the clinical setting.

Mori et al. measured the BRCAness of surgical specimens from 262 patients with primary TNBC who had not received neoadjuvant chemotherapy [22]. BRCAness in patients with TNBC was an independent factor for both recurrence and survival in their multivariate analysis. In the Kaplan–Meier analysis, however, there were no significant differences in RFS and OS between the BRCAness and non-BRCAness groups in patients with adjuvant chemotherapy. Our results of CNB specimens also showed similar results for RFS and OS in a Kaplan–Meier analysis. Therefore, the BRCAness of surgical specimens after neoadjuvant chemotherapy may be an important marker for predicting prognosis.

Most patients with TNBC who do not achieve pCR after neoadjuvant chemotherapy have a poor prognosis [7,8]. In contrast, patients with TNBC who achieve pCR have a good prognosis. In this study, patients who achieved pCR did not exhibit breast cancer recurrence (Table 4). Recently, the CREATE-X trial was found to prolong disease-free survival and OS, particularly in patients with TNBC [19]. However, it is still unclear whether adjuvant capecitabine should be administered to all patients because there are no biomarkers for recurrence after neoadjuvant chemotherapy.

Tutt et al. compared the efficacy of carboplatin with docetaxel in a phase III trial (TNT trial) with advanced TNBC [23]. In the trial, carboplatin induced a better objective response rate to germline *BRCA1/2* TNBC than docetaxel. The progression-free survival was also longer in patients with germline *BRCA1/2* mutations who were treated with carboplatin. Moreover, Telli et al. previously also reported that platinum-based agents were effective against tumors with a BRCAness profile [24]. Additional therapies for non-pCR patients after neoadjuvant therapies are needed, such as those evaluated in the CREATE-X trial of patients with TNBC and in the KATHERINE trial. Unfortunately, platinum is not available for triple negative breast cancer patients because of Japanese insurance. Therefore, we are recruiting patients who have not achieved pCR after neoadjuvant chemotherapy for a new clinical trial of adjuvant carboplatin with BRCAness (UMIN: 000030780).

There is a limitation that the correlations between germline *BRCA* mutations and BRCAness are not being evaluated in this study. Poly ADP ribose polymerase (PARP) inhibitors, which block DNA single-strand break repair, are beneficial for germline *BRCA*-mutated metastatic breast cancer [25,26,27,28]. A phase III trial (OlympiA) is currently ongoing to assess OS in patients with HER2-negative breast cancer with germline *BRCA* mutations treated with the PARP inhibitor Olaparib in the adjuvant setting (NCT02032823). A phase II trial (GEICAM/2015-06, COMETA-Breast) is also ongoing to evaluate the efficacy of Olaparib in patients with advanced TNBC with *BRCA1/2* promotor methylation (NCT03205761). In a recent experimental study, PARP inhibitor modulated cancer-associated immunosuppression by upregulating programmed death ligand 1 (PD-L1) in breast cancer cell lines suggesting that blockade of PD-L1 could restore their sensitivity to the PARP inhibitor. A subsequent xenograft study combining a PARP inhibitor to a PD-L1 inhibitor revealed a significant synergistic effect compared with either agent alone [29]. Accordingly, the feasibility of a combination treatment with a PD-L1 inhibitor and a PARP inhibitor is under exploration in the current phase II trial [30]. BRCAness has been investigated as a new therapeutic strategy through its pharmacological induction. Intriguing results suggest that the induction of a *BRCA*-mutant-like phenotype could be achieved through the epigenetic silencing of *BRCA1*, enhancing platinum salts’ activity and enabling the use of targeted drugs such as PARP inhibitors [31,32]. Based on these trials, we also expect that PARP inhibitors may be key drugs for BRCAness in patients with TNBC after neoadjuvant chemotherapy.

## 5. Conclusions

The BRCAness of surgical specimens from patients with TNBC after neoadjuvant chemotherapy was related to poor RFS or OS, and the BRCAness of CNB specimens may be predictive of clinical response to neoadjuvant chemotherapy. Different treatment approaches are needed to improve outcomes in patients with TNBC showing BRCAness who do not achieve pCR.

## Figures and Tables

**Figure 1 diagnostics-10-00119-f001:**
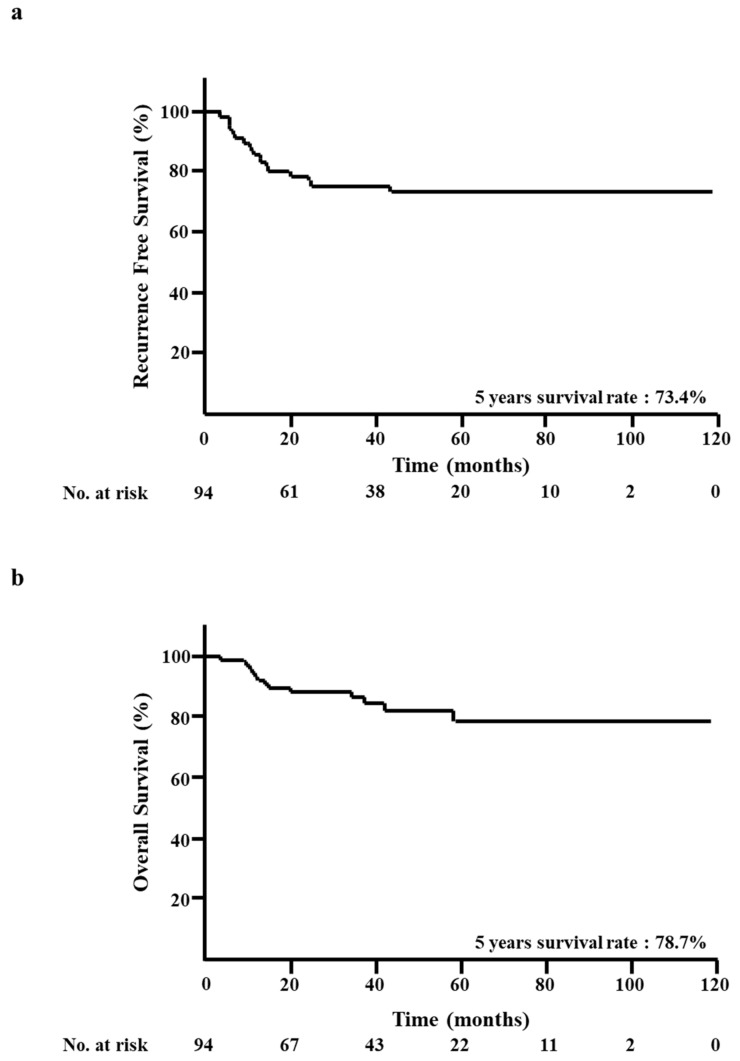
Kaplan–Meier analysis of all patients. (**a**) Recurrence-free survival (RFS). (**b**) Overall survival (OS).

**Figure 2 diagnostics-10-00119-f002:**
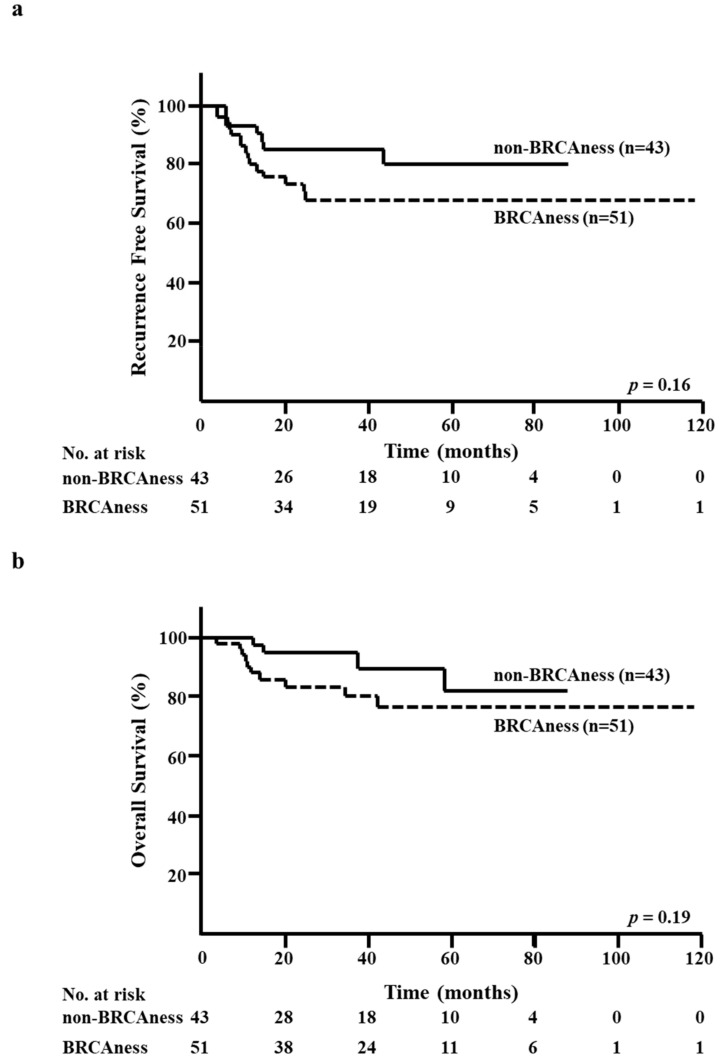
Kaplan–Meier analysis according to the BRCAness of core needle biopsy (CNB) specimens. (**a**) Recurrence-free survival (RFS) in the BRCAness and non-BRCAness groups. (**b**) Overall survival (OS) in the BRCAness and non-BRCAness groups.

**Figure 3 diagnostics-10-00119-f003:**
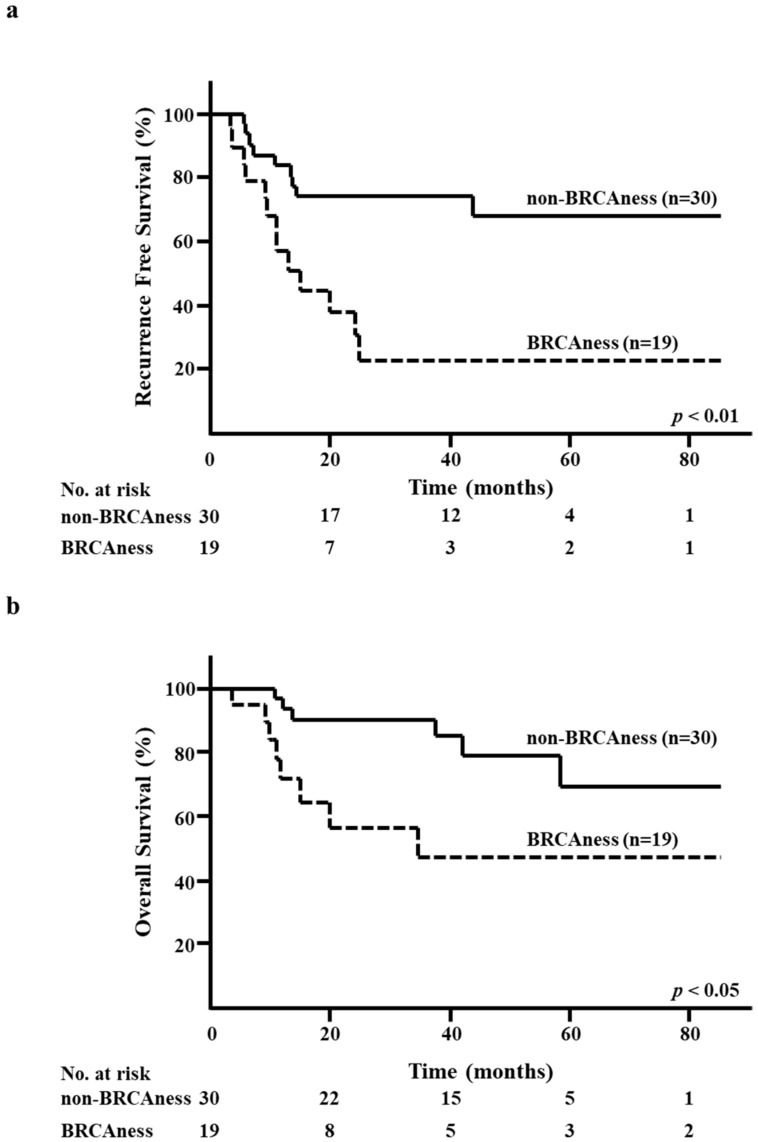
Kaplan–Meier analysis according to the BRCAness of surgical specimens after neoadjuvant chemotherapy. (**a**) Recurrence-free survival (RFS) in the BRCAness and non-BRCAness groups. (**b**) Overall survival (OS) in the BRCAness and non-BRCAness groups.

**Table 1 diagnostics-10-00119-t001:** Clinicopathologic characteristics of the 94 patients.

Factors	No.	%
Patient	94	100
Age (mean ± SD)	51.3 ± 11.4	
**cT**	**No.**	**%**
T1	10	10.6
T2	62	66
T3	9	9.6
T4	13	13.8
cN	No.	%
N0	33	35.1
N1	53	56.4
N2	6	6.4
N3	2	2.1
**cStage**	**No.**	**%**
I	7	7.4
II	64	68.1
III	23	24.5
**Neoadjuvant chemotherapy**	**No.**	**%**
Anthracycline followed by taxane	86	91.5
Anthracycline alone	3	3.2
Taxane alone	5	5.3
**Pathological complete response**	**No.**	**%**
No	49	52.1
Yes	45	47.9

**Table 2 diagnostics-10-00119-t002:** Correlation of clinicopathologic characteristics and BRCAness of biopsy.

Factors	Total	non-BRCAness		BRCAness		*p*
		(*n* = 43)		(*n* = 51)		
		No.	%	No.	%	
Age (mean ± SD)		51.2 ± 11.2		51.4 ± 11.6		NS
**Tumor size**	**Total**	**No.**	**%**	**No.**	**%**	***p***
cT1-cT2	72	33	76.7	39	76.5	NS
cT3-cT4	22	10	23.3	12	23.5	
**Lymph node metastasis**	**Total**	**No.**	**%**	**No.**	**%**	***p***
Negative (cN0)	33	14	32.6	19	37.3	NS
Positive (cN1-cN3)	61	29	67.4	32	62.7	
**cStage**	**Total**	**No.**	**%**	**No.**	**%**	***p***
I	7	4	9.3	3	5.9	NS
II	64	29	67.4	35	68.6	
III	23	10	23.3	13	25.5	
**Response**	**Total**	**No.**	**%**	**No.**	**%**	***p***
pCR	45	25	58.1	20	39.2	NS
non-pCR	49	18	41.9	31	60.8	

NS: not significant.

**Table 3 diagnostics-10-00119-t003:** Correlations of clinicopathologic characteristics and BRCAness in surgical specimens.

Factors	Total	non-BRCAness		BRCAness		*p*
		(*n* = 30)		(*n* = 19)		
		No.	%	No.	%	
Age (mean ± SD)		53.5 ± 10.4		47.0 ± 11.6		< 0.05
**Tumor size**	**Total**	**No.**	**%**	**No.**	**%**	***p***
cT1-cT2	35	22	73.3	13	68.4	NS
cT3-cT4	14	8	26.7	6	31.6	
**Lymph node metastasis**	**Total**	**No.**	**%**	**No.**	**%**	***p***
Negative (cN0)	15	9	30.0	6	31.6	NS
Positive (cN1-cN3)	34	21	70.0	13	68.4	
**cStage**	**Total**	**No.**	**%**	**No.**	**%**	***p***
I	2	2	6.7	0	0.0	NS
II	32	19	63.3	13	68.4	
III	15	9	30.0	6	31.6	
**Recurrence**	**Total**	**No.**	**%**	**No.**	**%**	***p***
No	27	21	70.0	6	31.6	< 0.05
Yes	22	9	30.0	13	68.4	

NS: not significant.

**Table 4 diagnostics-10-00119-t004:** Correlation of clinicopathological characteristics and pCR.

Factors	Total	non-pCR		pCR		*p*
		(*n* = 49)		(*n* = 45)		
		No.	%	No.	%	
Age (mean ± SD)		51.0 ± 11.2		51.6 ± 11.7		NS
**Tumor size**	**Total**	**No.**	**%**	**No.**	**%**	***p***
cT1-cT2	72	35	71.4	37	82.2	NS
cT3-cT4	22	14	28.6	8	17.8	
**Lymph node metastasis**	**Total**	**No.**	**%**	**No.**	**%**	***p***
Negative (cN0)	33	15	30.6	18	40.0	NS
Positive (cN1-3)	61	34	69.4	27	60.0	
**Stage**	**Total**	**No.**	**%**	**No.**	**%**	***p***
I	7	2	4.1	5	11.1	NS
II	64	32	65.3	32	71.1	
III	23	15	30.6	8	17.8	
**Recurrence**	**Total**	**No.**	**%**	**No.**	**%**	***p***
No	72	27	55.1	45	100.0	< 0.0001
Yes	22	22	44.9	0	0.0	
**Overall survival**	**Total**	**No.**	**%**	**No.**	**%**	***p***
Alive	80	35	71.4	45	100.0	< 0.0001
Dead	14	14	28.6	0	0.0	

NS: not significant.

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
