# Peer review of "BRCAness as an Important Prognostic Marker in Patients with Triple-Negative Breast Cancer Treated with Neoadjuvant Chemotherapy: A Multicenter Retrospective Study"

_diagnostics, 2020, doi:10.3390/diagnostics10020119_

Round 1

Reviewer 1 Report

The work evaluated the prognostic role of BRCAness in a retrospective series of patients with triple negative breast cancer treated with neoadjuvant chemotherapy.

The small sample size and the retrospective design of the study make results less reliable. 

Prognosis should be evaluates also on the entire series of 94 cases and not limited to patients with residual disease after neoadjuvant chemotherapy

Information about chemotherapy should be more precise. Regimens must be described in terms of doses, number of courses, duration. In addition, the nonuse of platinum salts should be explained and commented.

Relevant papers were not quoted. They should be commented in the discussion and added to the list of references:  

1: Garutti M, et al. Platinum Salts in Patients with Breast Cancer: A Focus on Predictive Factors. Int J Mol Sci 2019;20(14). 

2: Mio C, et al. BET proteins regulate homologous recombination-mediated DNA repair: BRCAness and implications for cancer therapy.  Int J Cancer 2019;144(4):755-766. 

Author Response

Point 1: Prognosis should be evaluates also on the entire series of 94 cases and not limited to patients with residual disease after neoadjuvant chemotherapy

Response 1: We added the RFS and OS of all patients as Figure 1 at results section.

Point 2: Information about chemotherapy should be more precise. Regimens must be described in terms of doses, number of courses, duration. In addition, the nonuse of platinum salts should be explained and commented.

Response 2: We added the information about chemotherapy regimens precisely at patients section. And the reason of nonuse of platinum salt was mentioned at patients and discussion section. Actually, it is not covered by Japanese insurance for triple negative breast cancer.

Point 3: Relevant papers were not quoted. They should be commented in the discussion and added to the list of references: 

1: Garutti M, et al. Platinum Salts in Patients with Breast Cancer: A Focus on Predictive Factors. Int J Mol Sci 2019;20(14).

2: Mio C, et al. BET proteins regulate homologous recombination-mediated DNA repair: BRCAness and implications for cancer therapy.  Int J Cancer 2019;144(4):755-766.

Response 3: We mentioned the papers as reference 31 and 32 at discussion section.

Reviewer 2 Report

Well done!

Please suggest somewhere in the discussions the type of novel treatments that could be available to those patients in future that you might improve the prognosis (e.g. immunotherapies or targeting p53 mutant antigen)

Author Response

Point: Please suggest somewhere in the discussions the type of novel treatments that could be available to those patients in future that you might improve the prognosis (e.g. immunotherapies or targeting p53 mutant antigen)

Response: We mentioned about immunotherapy and PARP inhibitor synergistic effect at discussion section. There was no evidence as Phase III trial. However, it might be novel treatments for the patients.

Round 2

Reviewer 1 Report

No other revisions are requested.